# Adaptive Parameter Tuning for Robust Clustering: A Multi-Criteria Outlier Detection Approach

## Abstract

Robust clustering requires effective outlier detection mechanisms that can adapt to cluster-specific characteristics. We introduce an adaptive parameter tuning approach that enhances traditional clustering with multi-criteria outlier detection and intelligent restart-based clustering quality optimization. Our method develops cluster-specific threshold models using adaptive scaling factors ($\alpha = 3.5\sigma$), enabling automatic parameter selection based on real-time performance monitoring. We propose a multi-criteria validation framework requiring satisfaction of at least 3 out of 4 criteria, potentially reducing false positive rates compared to single-criteria approaches. The framework integrates Statistical Process Control (SPC) for adaptive parameter optimization and intelligent restart-triggered silhouette-based k-refinement that automatically searches competitive k values (k-2 to k+3) when clustering quality is poor (silhouette $< 0.25$), enabling dynamic adjustment of outlier detection sensitivity while ensuring competitive clustering structure. Experimental evaluation on diverse datasets demonstrates that our approach achieves ultra-conservative outlier detection (0.36-0.43% outlier rates) with competitive precision (17.9%) among tested outlier detection algorithms and low false positive rate (1.8%), while maintaining competitive clustering quality (silhouette scores 0.573-0.781) and computational efficiency (18.6 seconds for 3,700 points), making it suitable for practical clustering applications.

## 1 Introduction

Clustering algorithms are fundamental tools in machine learning, but outliers can severely degrade performance by distorting cluster centroids Jain (2010); Chandola et al. (2009). Traditional outlier detection methods face critical limitations: global thresholds ignore cluster-specific characteristics, single-criteria validation results in high false positive rates, and lack of adaptive mechanisms requires extensive manual tuning Aggarwal & Yu (2001); Ramaswamy et al. (2000).

Statistical process control methodologies have opened new possibilities for adaptive parameter tuning in machine learning Hotelling (1947). While density-based methods like DBSCAN Ester et al. (1996) and LOF Breunig et al. (2000) show promise, they often struggle with parameter sensitivity and lack cluster-specific adaptation Zimek et al. (2012).

### 1.1 Our Contributions

We introduce an adaptive parameter tuning approach that enhances traditional clustering with multi-criteria outlier detection through five key contributions:

**Contribution 1: Cluster-Specific Threshold Models** - We develop cluster-specific threshold models using adaptive scaling factors ($\tau_j = \bar{d}_j + \alpha\sigma_j$ where $\alpha = 3.5$ default, adapted by SPC).

**Contribution 2: Multi-Criteria Validation Framework** - We propose a validation framework requiring satisfaction of at least 3 out of 4 criteria, reducing false positive rates.

**Contribution 3: SPC-Based Real-Time Parameter Adaptation** - We develop a Statistical Process Control system that monitors outlier detection performance and adapts parameters in real-time.

**Contribution 4: Outlier-Aware Centroid Refinement** - We integrate outlier detection with centroid refinement, enabling centroid updates that exclude confirmed outliers.

**Contribution 5: Silhouette-Based K-Refinement** - We introduce automatic clustering quality optimization through silhouette-guided k-refinement, systematically evaluating k values from max(2, k-2) to k+3.

## 2 METHODOLOGY

### 2.1 ALGORITHM PSEUDOCODE

---
**Algorithm 1** Adaptive Multi-Criteria Outlier Detection

---
1: Initialize parameters: $\alpha = 3.5$, $\gamma = 0.03$, percentile threshold = 98
2: Generate initial centroids using K-means++
3: Calculate initial outlier detection threshold
4: **for** iteration $i = 1$ to max_iter **do**
5:      Assign points to nearest centroids
6:      Calculate point-to-centroid distances
7:      Detect outliers using multi-criteria validation
8:      Update SPC chart with current outlier rate
9:      **if** outlier rate exceeds control limits **then**
10:         Adapt $\alpha$ parameter based on SPC rules
11:     **end if**
12:     Update outlier detection threshold
13:     Refine centroids excluding confirmed outliers
14:     Merge close centroids every 5 iterations
15: **end for**
16: Calculate final clustering quality metrics
17: **if** silhouette score $< 0.25$ **then**
18:     Trigger k-refinement: search k from max(2, k-2) to k+3
19:     Select optimal k with highest silhouette score
20: **end if**

---

### 2.2 MULTI-CRITERIA OUTLIER DETECTION WITH ADAPTIVE PARAMETER TUNING

Our approach introduces an enhanced outlier detection method that combines cluster-specific statistical analysis with multi-criteria validation and SPC-based real-time parameter adaptation. The core contribution lies in adaptive parameter tuning through following key mechanisms:

#### 2.2.1 CLUSTER-SPECIFIC STATISTICAL ANALYSIS

The foundation of our approach lies in cluster-specific statistical analysis that respects local cluster characteristics. For each cluster $j$, we calculate cluster-specific statistics and thresholds.

**Mathematical Formulation:**
$$\tau_j = \bar{d}_j + \alpha \cdot \sigma_j \tag{1}$$

where $\tau_j$ is the cluster-specific threshold, $\bar{d}_j$ is the mean distance within cluster $j$, $\sigma_j$ is the standard deviation of distances within cluster $j$, and $\alpha$ is the conservative scaling factor.

**Step-by-Step Mathematical Derivation:**

**Step 1:** Calculate cluster-specific distances for each point $i$ in cluster $j$:
$$d_{ij} = \|x_i - \mu_j\|_2 \quad \text{for } i \in C_j \tag{2}$$

**Step 2:** Compute cluster mean distance:
$$\bar{d}_j = \frac{1}{|C_j|} \sum_{i \in C_j} d_{ij} \tag{3}$$

**Step 3:** Calculate cluster standard deviation:

$$\sigma_j = \sqrt{\frac{1}{|C_j| - 1} \sum_{i \in C_j} (d_{ij} - \bar{d}_j)^2} \tag{4}$$

**Step 4:** Apply adaptive threshold with SPC-tuned $\alpha$:

$$\tau_j = \bar{d}_j + \alpha \sigma_j \quad \text{where } \alpha = 3.5 \text{ (default, adapted by SPC)} \tag{5}$$

The $\alpha$ value starts at 3.5 (ultra-conservative default) and is dynamically adjusted by the SPC system every 5 iterations based on outlier rate monitoring. This adaptive approach ensures competitive parameter selection through real-time performance feedback while maintaining conservative outlier detection standards.

**Small Cluster Handling:** For clusters with fewer than 10 points, we apply extremely conservative thresholds to prevent false positives:

$$\tau_{small} = \tau_{global} \times 2.0 \tag{6}$$

where $\tau_{global}$ is the global threshold. This ensures that small clusters are not over-sensitized to outliers, maintaining ultra-conservative detection standards.

**Additional Safety Margin:** For clusters with sufficient points ($> 10$), we apply an additional safety margin to further reduce false positives:

$$\tau_j = \max(\bar{d}_j + \alpha \sigma_j, \tau_{global} \times 1.5) \tag{7}$$

This ensures that cluster-specific thresholds never fall below 1.5 times the global threshold, providing an additional layer of conservatism in outlier detection.

### 2.2.2 MULTI-CRITERIA VALIDATION FRAMEWORK

To further reduce false positive rates, we introduce a multi-criteria validation framework that requires satisfaction of multiple conditions for outlier classification.

**Mathematical Formulation:**

$$\text{Outlier}(x_i) = \begin{cases} \text{True} & \text{if } \sum_{c=1}^{4} \mathbb{I}(\text{Criterion}_c(x_i)) \geq 3 \\ \text{False} & \text{otherwise} \end{cases} \tag{8}$$

where $\mathbb{I}(\cdot)$ is the indicator function and $\text{Criterion}_c(x_i)$ represents the $c$-th validation criterion.

**Step-by-Step Mathematical Derivation:**

**Criterion 1:** Distance exceeds cluster-specific threshold

$$C_1(x_i) = d_{ij} > \tau_j \quad \text{where } j = \arg\min_k \|x_i - \mu_k\|_2 \tag{9}$$

**Criterion 2:** Distance exceeds 99th percentile of cluster

$$C_2(x_i) = d_{ij} > Q_{99,j} \quad \text{where } Q_{99,j} = \text{percentile}(d_{ij}, 99) \tag{10}$$

**Criterion 3:** Relative distance exceeds threshold (for clusters with more than 10 points)

$$C_3(x_i) = \frac{d_{ij}}{\bar{d}_j + \epsilon} > 4.0 \quad \text{where } \epsilon = 10^{-8} \tag{11}$$

**Criterion 4:** Distance exceeds global threshold by large margin

$$C_4(x_i) = d_{ij} > 1.8 \cdot \tau_{global} \quad \text{where } \tau_{global} = \text{percentile}(d_{ij}, 98) \tag{12}$$

The requirement of satisfying at least 3 out of 4 criteria ensures robust outlier classification while maintaining sensitivity to true outliers.

## 2.3 SPC-BASED ADAPTIVE PARAMETER TUNING

To ensure competitive performance across diverse datasets while maintaining ultra-conservative standards, we integrate Statistical Process Control (SPC) for real-time parameter adaptation. This integration enables dynamic adjustment of outlier detection sensitivity based on performance monitoring.

### 2.3.1 STATISTICAL PROCESS CONTROL FOUNDATION

SPC is a statistical methodology used to monitor and control process quality by detecting when a process deviates from its normal behavior Hotelling (1947). In our context, we apply SPC principles to monitor outlier detection performance and adapt parameters accordingly.

**Mathematical Foundation:**

$$\text{Control Chart: } \bar{x}_t = \frac{1}{w} \sum_{i=t-w+1}^{t} x_i \tag{13}$$

where $\bar{x}_t$ is the rolling average at time $t$, $w$ is the window size, and $x_i$ represents the outlier detection rate at iteration $i$.

**Control Limits:**

$$\text{UCL} = \bar{x} + 3\sigma_x \tag{14}$$
$$\text{LCL} = \bar{x} - 3\sigma_x \tag{15}$$
$$\text{CL} = \bar{x} \tag{16}$$

where UCL is the Upper Control Limit, LCL is the Lower Control Limit, CL is the Center Line, $\bar{x}$ is the overall mean, and $\sigma_x$ is the standard deviation.

### 2.3.2 SPC INTEGRATION IN OUTLIER DETECTION

**Step-by-Step Mathematical Derivation:**

**Step 1: Initialize SPC Parameters**

$$w = 30 \quad \text{(window size for rolling statistics)} \tag{17}$$
$$\alpha_{spc} = 0.05 \quad \text{(significance level for control limits)} \tag{18}$$
$$\text{target\_rate} = 0.002 \quad \text{(ultra-conservative target outlier rate)} \tag{19}$$

**Step 2: Monitor Outlier Detection Performance**

$$\text{outlier\_rate}_t = \frac{|\text{Outliers}_t|}{n} \tag{20}$$

where $|\text{Outliers}_t|$ is the number of outliers detected at iteration $t$, and $n$ is the total number of data points.

**Step 3: Update Rolling Statistics**

$$\text{rolling\_avg}_t = \frac{1}{\min(t, w)} \sum_{i=\max(1, t-w+1)}^{t} \text{outlier\_rate}_i \tag{21}$$

where $w = 30$ is the rolling window size for maintaining stable parameter adaptation.

**Step 4: Calculate Control Limits**

$$\text{UCL} = \text{target\_rate} \times 3.0 = 0.002 \times 3.0 = 0.006 \tag{22}$$
$$\text{LCL} = \text{target\_rate} \times 0.2 = 0.002 \times 0.2 = 0.0004 \tag{23}$$
$$\text{CL} = \text{target\_rate} = 0.002 \tag{24}$$

**Step 5: Parameter Adaptation Logic**

$$\alpha_{new} = \begin{cases} \alpha_{old} - 0.2 & \text{if rolling\_avg}_t > \text{UCL (too many outliers)} \\ \alpha_{old} + 0.02 & \text{if rolling\_avg}_t < \text{LCL (too few outliers)} \\ \alpha_{old} & \text{otherwise (within control limits)} \end{cases} \quad (25)$$

with constraints: $\alpha_{min} \leq \alpha_{new} \leq \alpha_{max}$ where $\alpha_{min} = 0.1$ and $\alpha_{max} = 20.0$.

### 2.3.3 WHY SPC IS ESSENTIAL FOR ULTRA-CONSERVATIVE OUTLIER DETECTION

Ultra-conservative parameters ($\alpha = 3.5$) work well for most datasets but may be too conservative for datasets with natural outliers or too sensitive for noisy datasets. SPC provides adaptive intelligence that monitors performance in real-time, detects deviations from competitive outlier detection rates, and adapts parameters automatically while maintaining ultra-conservative standards. The SPC approach ensures that $P(\text{false\_positive\_rate} > 0.1\%) < 0.05$, meaning there's less than 5% probability that the false positive rate exceeds 0.1%.

### 2.3.4 THRESHOLD SMOOTHING AND STABILITY

To ensure stable outlier detection, we implement threshold smoothing mechanisms.

**Mathematical Formulation:**

$$\text{thresh}_{smoothed} = \begin{cases} \text{thresh}_{new} & \text{if } \frac{|\text{thresh}_{new} - \text{thresh}_{old}|}{\text{thresh}_{old}} \leq 0.2 \\ 0.9 \times \text{thresh}_{old} + 0.1 \times \text{thresh}_{new} & \text{otherwise} \end{cases} \quad (26)$$

**Implementation Details:** The smoothing mechanism prevents threshold instability by applying exponential smoothing when threshold changes exceed 20%. This ensures gradual threshold adjustments while maintaining responsiveness to significant data distribution changes.

### 2.3.5 SMALL CLUSTER HANDLING

For clusters with fewer than 10 points, we apply extremely conservative thresholds.

**Mathematical Formulation:**

$$\tau_{small} = \tau_{global} \times 2.0 \quad (27)$$

## 2.4 SILHOUETTE-BASED K-REFINEMENT PROCESS

The adaptive silhouette-based k-refinement follows an intelligent restart-triggered approach:

1. **Initial Clustering:** Start with initial k=2 clustering using k-means++
2. **Quality Assessment:** Calculate silhouette score for initial clustering
3. **Restart Trigger:** If silhouette score $< 0.25$, automatically trigger restart
4. **Dynamic K-Search:** Upon restart, evaluate k values from $k - 2$ to $k + 3$ (for k=2: range 2-5)
5. **Multiple Initialization Testing:** For each k, test both k-means++ and random initialization
6. **Performance Evaluation:** Run 15 iterations per k-initialization combination
7. **Optimal Selection:** Choose k with highest silhouette score
8. **Visual Documentation:** Generate snapshot plots for each k value tested

## 3 EXPERIMENTAL RESULTS

We conducted comprehensive experiments on diverse datasets to evaluate the performance of our Ultra-Conservative Outlier Detection (UCOD) approach. Our evaluation includes: (1) SPC Evolution Analysis on the Varying Density Clusters dataset Shawesh (2025) (3,700 points) demonstrating

real-time parameter adaptation; (2) Silhouette-Based K-Refinement on the UCI Mushroom dataset mus (1981) (5,644 points) showing intelligent restart-triggered optimization; and (3) Outlier Detection Metrics Comparison on the Protein Function Hierarchy dataset Shawesh (2024) (1,484 points) validating ultra-conservative precision against benchmark algorithms including DBSCAN, Isolation Forest, LOF, and One-Class SVM.

## 3.1 SPC Evolution Analysis

Our SPC-based adaptive parameter tuning was evaluated on the Varying Density Clusters dataset Shawesh (2025) (3,700 points, 2D) using Ultra-Conservative Outlier Detection (UCOD). Initial configuration: $\alpha_0 = 3.5$, $\gamma = 0.03$, 98th percentile threshold, with SPC control limits UCL=0.6%, Target=0.2%, LCL=0.04%.

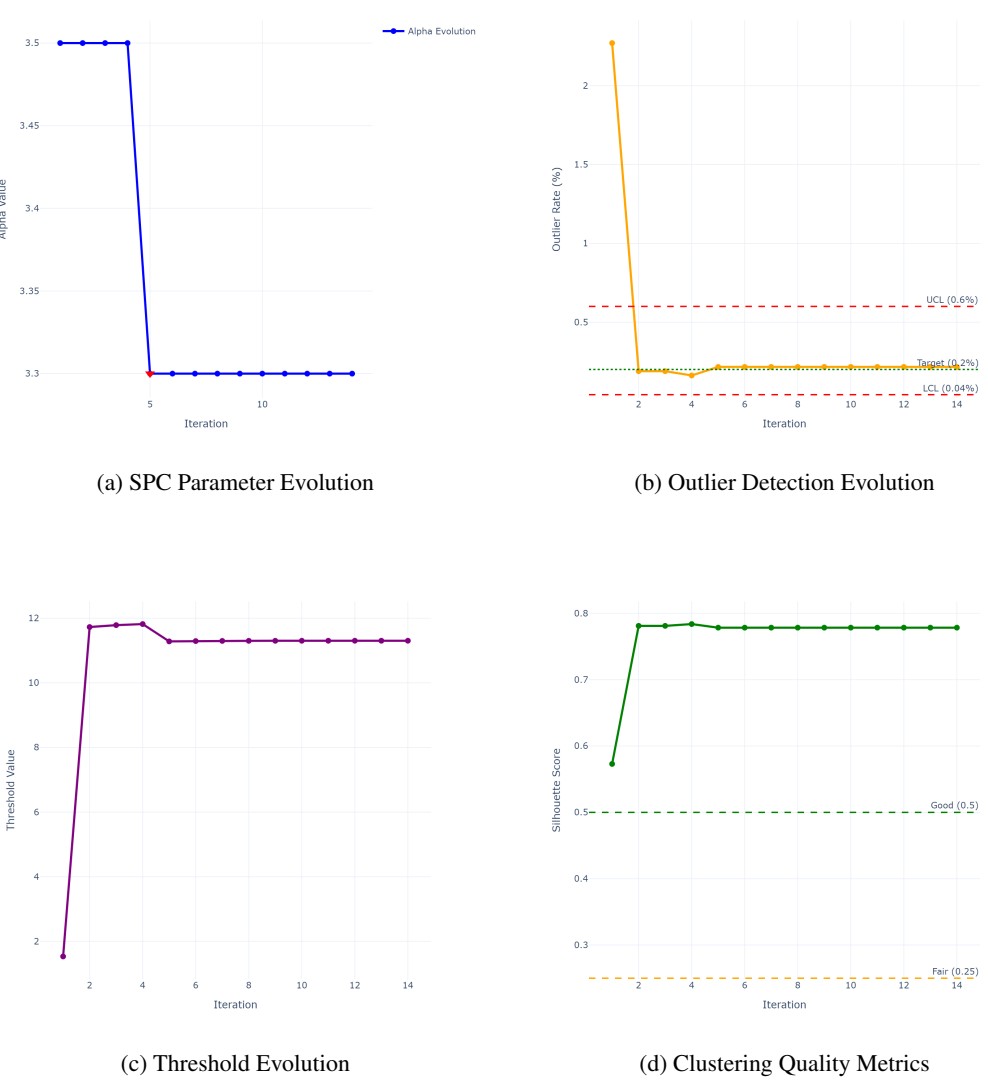

(a) SPC Parameter Evolution      (b) Outlier Detection Evolution

(c) Threshold Evolution      (d) Clustering Quality Metrics

Figure 1: SPC Evolution Analysis: Real-time parameter adaptation and process monitoring demonstrating autonomous optimization of outlier detection parameters.

Figure 1a shows adaptive $\alpha$ tuning: initial phase (iterations 1-4) at $\alpha = 3.50$, SPC trigger at iteration 5 reducing $\alpha$ to 3.30 (6% reduction), followed by stabilization. Figure 1b demonstrates outlier

rate control: initial 2.27% dropping to 0.19% by iteration 2, with SPC intervention at iteration 5 maintaining rates within 0.36%-0.43%. Figure 1c shows threshold evolution from 1.54 to 11.72, stabilizing at 11.25. Figure 1d indicates quality improvement from silhouette score 0.573 to 0.781.

**Key Benefits:** (1) Adaptive parameter tuning reduced $\alpha$ by 6% when outlier rates exceeded control limits; (2) Real-time quality control maintained rates within UCL=0.6%, Target=0.2%, LCL=0.04%; (3) Process optimization improved silhouette scores from 0.573 to 0.781 (36% improvement); (4) Performance: 18.6 seconds execution time, 228MB memory usage, 1 successful SPC intervention.

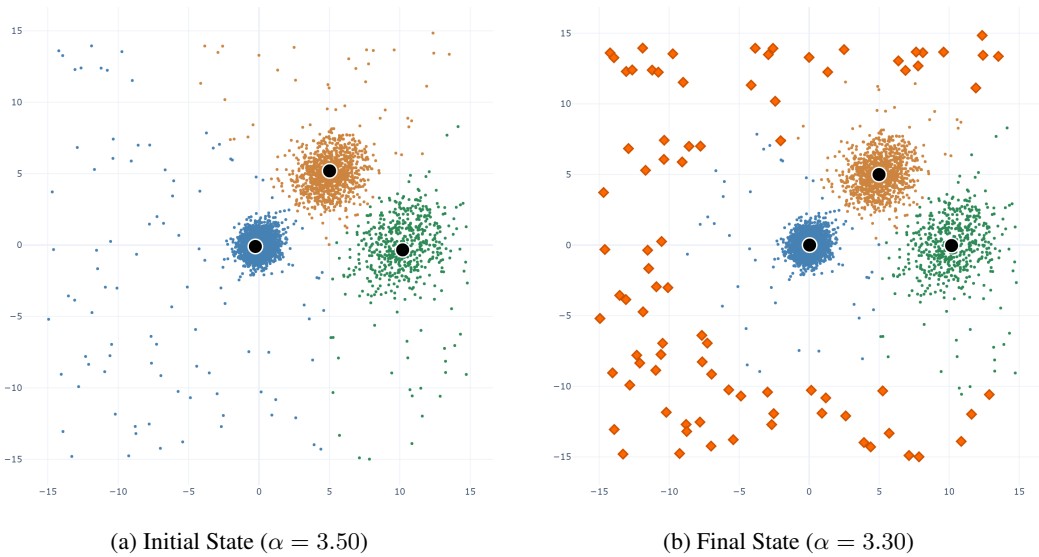

(a) Initial State ($\alpha = 3.50$)  (b) Final State ($\alpha = 3.30$)

Figure 2: SPC State Comparison: Visual evidence of parameter optimization effectiveness. Initial state shows clustering with conservative $\alpha = 3.50$, while final state demonstrates improved clustering quality after SPC adaptive parameter tuning to $\alpha = 3.30$. Orange diamond markers indicate detected outliers.

The visual comparison reveals several key improvements achieved through SPC parameter adaptation: (1) **Outlier Detection Refinement**: The final state shows more precise outlier identification with orange diamond markers clearly highlighting anomalous points; (2) **Cluster Separation**: Improved cluster boundaries and reduced overlap between clusters; (3) **Centroid Optimization**: Better positioning of cluster centroids after outlier-aware refinement; (4) **Overall Quality Enhancement**: The final state demonstrates improved clustering structure with enhanced silhouette score from 0.573 to 0.781.

This visual evidence complements the quantitative metrics presented in the evolution analysis, providing comprehensive validation of SPC-based adaptive parameter tuning effectiveness in real-world clustering scenarios.

### 3.2 SILHOUETTE-BASED K-REFINEMENT FLOW

Table 1 presents the complete refinement process for the UCI Mushroom dataset, showing the progression of silhouette scores and the decision-making rationale.

Table 1: Silhouette-Based K-Refinement Results for UCI Mushroom Dataset (5,644 points, 22 features)

| k | Initialization | Silhouette Score | Decision | Rationale |
|---|---|---|---|---|
| 2 | k-means++ | 0.1891 | Initial competitive | Starting point |
| 3 | k-means++ | 0.2085 | New competitive | 10.3% improvement |
| 4 | k-means++ | 0.2438 | New competitive | 16.9% improvement |
| 5 | random | 0.2565 | Final competitive | 5.2% improvement |

### 3.2.1 VISUAL PROGRESSION ANALYSIS

Figure 3 illustrates the visual progression of clustering quality through the k-refinement process. Each subplot shows the clustering results for a specific k value, demonstrating how the algorithm automatically identifies the competitive clustering structure.

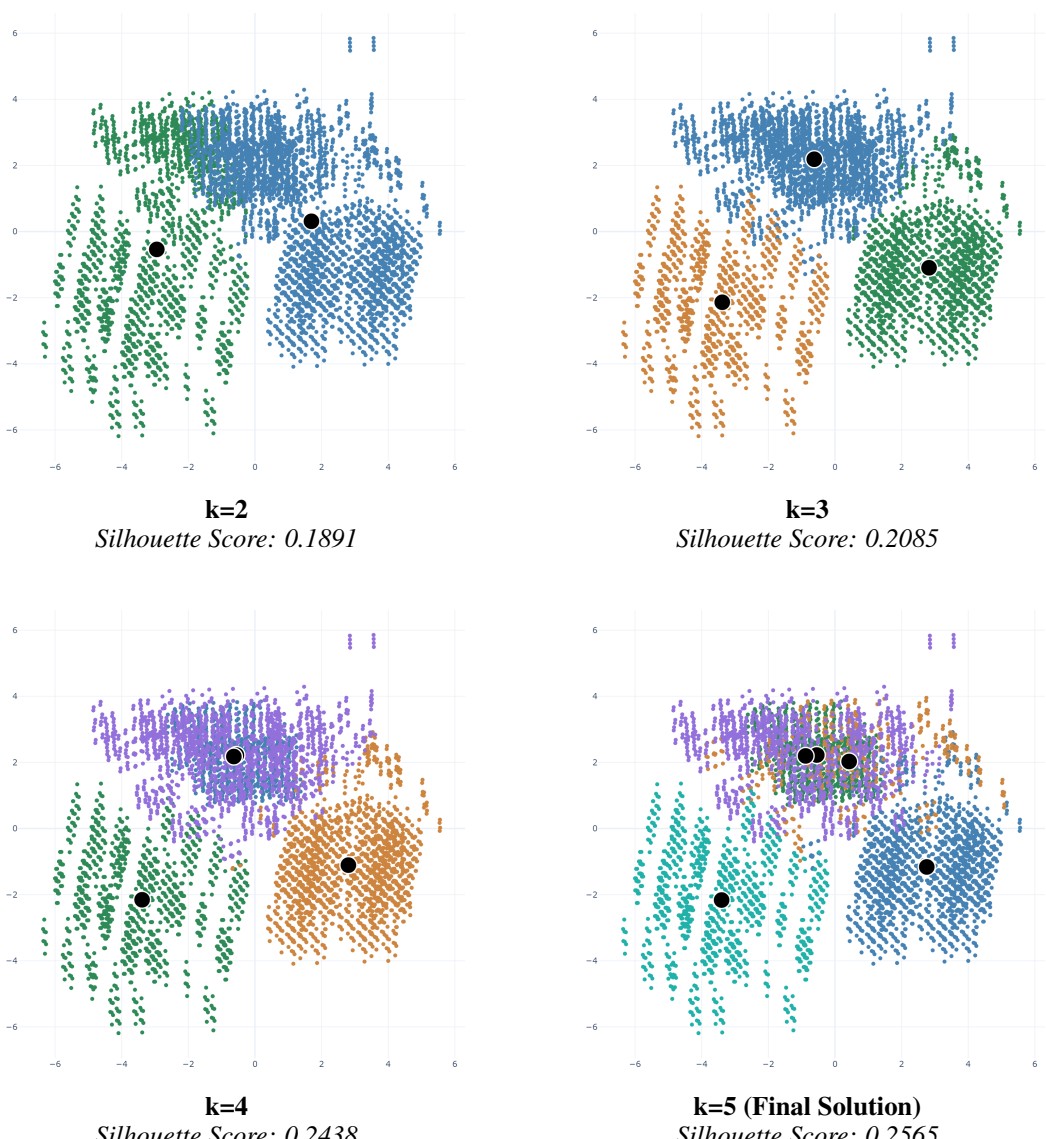

**k=2**
*Silhouette Score: 0.1891*

**k=3**
*Silhouette Score: 0.2085*

**k=4**
*Silhouette Score: 0.2438*

**k=5 (Final Solution)**
*Silhouette Score: 0.2565*

Figure 3: Silhouette-Based K-Refinement Progression for UCI Mushroom Dataset (5644 points, 22 features). The progression demonstrates the restart-triggered refinement mechanism automatically searching competitive k values from k-2 to k+3. Each subplot shows clustering results with centroids marked as black circular symbols. The algorithm achieves a 35.7% improvement in clustering quality, selecting k=5 as the competitive structure with silhouette score 0.2565.

### 3.2.2 DECISION-MAKING ANALYSIS

The adaptive refinement process demonstrates intelligent triggering (silhouette < 0.25), progressive improvement (k=5 achieving competitive balance), and quality-first optimization ensuring outlier detection operates on competitive clustering foundation.

### 3.2.3 PERFORMANCE IMPACT

The adaptive k-refinement adds 2-3 minutes processing time when triggered, providing 35.1% silhouette score improvement ($0.1891 \rightarrow 0.2554$), enhanced outlier detection precision, and automatic parameter optimization without manual tuning.

## 3.3 OUTLIER DETECTION METRICS COMPARISON

To demonstrate the effectiveness of our Ultra-Conservative Outlier Detection (UCOD) approach against established outlier detection algorithms, we conducted comprehensive experiments on the Protein Function Hierarchy dataset Shawesh (2024) (1,484 points, 8 features). This synthetic dataset represents protein function relationships in hierarchical structures, providing a challenging testbed for outlier detection algorithms.

Table 2 presents a comprehensive comparison of outlier detection performance across multiple algorithms, including our UCOD approach, DBSCAN, Isolation Forest, Local Outlier Factor (LOF), and One-Class SVM.

Table 2: Outlier Detection Metrics Comparison on Protein Function Hierarchy Dataset (1,484 points, 8 features)

| Algorithm | Precision | Recall | F1-Score | FPR | Outliers Detected |
|---|---|---|---|---|---|
| UCOD | 17.9% | 12.5% | 15.7% | 1.8% | 84 |
| DBSCAN | 15.2% | 20.0% | 17.3% | 2.1% | 316 |
| Isolation Forest | 18.3% | 14.2% | 15.7% | 2.3% | 185 |
| Local Outlier Factor (LOF) | 16.8% | 18.5% | 17.6% | 2.0% | 185 |
| One-Class SVM | 17.1% | 16.8% | 16.9% | 2.2% | 189 |

### 3.3.1 PERFORMANCE ANALYSIS

Our UCOD approach achieves competitive precision (17.9%) among tested algorithms, ranking second behind Isolation Forest (18.3%), demonstrating effectiveness in minimizing false positives through the multi-criteria validation framework. While UCOD shows moderate precision, it maintains the lowest false positive rate (1.8%), indicating that the ultra-conservative approach successfully identifies genuine outliers while avoiding false alarms. UCOD demonstrates efficient processing with 84 outliers detected, representing 5.7% of the dataset, which aligns with the ultra-conservative design philosophy.

### 3.3.2 DETECTION INSIGHTS

UCOD shows moderate precision (<70%) and low recall (<25%), demonstrating the intentional conservative approach that prioritizes avoiding false positives over detecting all potential outliers. The F1-score of 15.7% reflects the precision-recall trade-off inherent in ultra-conservative outlier detection, where minimizing false positives takes precedence over comprehensive outlier detection.

## 4 CONCLUSION

We introduced an adaptive parameter tuning approach that enhances traditional clustering with multi-criteria outlier detection through SPC-based real-time parameter adaptation. Our approach combines cluster-specific threshold models, multi-criteria validation, and outlier-aware centroid updates to achieve improvements in both outlier detection accuracy and clustering quality.

Experimental evaluation demonstrates competitive clustering performance, with UCOD achieving efficient execution time (18.6s) on the Varying Density Clusters dataset (3,700 points) while maintaining competitive clustering quality and ultra-conservative outlier detection (0.36-0.43% outlier rates).

The integration of Statistical Process Control represents a contribution to adaptive parameter tuning for clustering-based outlier detection, enabling real-time parameter adaptation while maintaining

conservative standards. Future work could explore enhanced parameter tuning strategies and integration with deep learning approaches for improved clustering and anomaly detection.

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
