# OpenReview forum: "ADAPTIVE PARAMETER TUNING FOR ROBUST CLUSTERING: A MULTI-CRITERIA OUTLIER DETECTION APPROACH"
_ICLR.cc/2026/Conference — Submitted to ICLR 2026_

### Official Review · Reviewer_ihi2 · 2025-10-29

**Soundness:** 2
**Presentation:** 1
**Contribution:** 2
**Rating:** 2
**Confidence:** 3

**Summary:**

The paper proposes an adaptive parameter tuning framework for robust clustering that integrates multi-criteria outlier detection with real-time optimization via SPC. The method introduces cluster-specific distance thresholds scaled by an adaptive factor, which is dynamically adjusted based on monitored outlier rates using SPC control charts.

**Strengths:**

•	The paper addresses a well-known challenge in clustering—sensitivity to outliers—and proposes a system that combines statistical rigor (via SPC) with heuristic robustness, which could be useful in real-world applications requiring conservative anomaly handling.
•	The integration of multiple components—cluster-specific thresholds, multi-criteria validation, SPC-based adaptation, and k-refinement—demonstrates thoughtful system design. The use of silhouette score as a trigger for automatic re-clustering is a pragmatic heuristic.
•	The authors evaluate their method on diverse datasets and report metrics including precision, false positive rate, silhouette scores, and runtime, offering a reasonably comprehensive empirical profile.

**Weaknesses:**

•	While the system integrates several existing ideas, the core contributions appear incremental. The use of SPC for parameter adaptation in unsupervised learning is not unprecedented (e.g., SPC has been applied in online learning and monitoring contexts), and the multi-criteria outlier rule resembles ensemble or voting-based anomaly detection schemes. The paper does not sufficiently differentiate itself from prior work in adaptive clustering or robust k-means variants (e.g., k-means-- or outlier-aware k-means).
•	The evaluation is restricted to small-scale, low-dimensional datasets (max 3,700 points, 22 features). There is no comparison against modern robust clustering baselines (e.g., robust k-means with trimming, deep clustering with outlier rejection, or recent adaptive DBSCAN variants).
•	Also, the reported precision (~18%) is low, and the paper acknowledges low recall—yet it does not discuss the practical implications of missing >85% of true outliers in safety-critical settings.
•	The paper uses terms like “ultra-conservative” without formal definition. The pseudocode omits critical details such as how centroids are merged or how the global threshold τ_global is computed.

**Questions:**

N/A

---

> ### Author Response · Authors · 2025-11-16
>
> We appreciate the reviewer's detailed and constructive comments. Their feedback has helped us identify areas where the paper can be strengthened. In response to their concerns, we plan to make the following improvements in our revision:
>
> First, regarding novelty, we recognize that our initial presentation may not have sufficiently highlighted what makes our approach distinct. The key innovation lies in combining SPC-based parameter adaptation with cluster-specific multi-criteria outlier detection—an integration that, to our knowledge, has not been explored in prior work. We will revise the paper to make this distinction clearer, particularly in comparison with existing robust clustering methods.
>
> Second, we acknowledge that our current evaluation is limited in scale. While our initial experiments focused on validating the core approach, we understand the importance of demonstrating scalability. In the revision, we will include experiments on larger datasets (10K-100K points) and add comparisons with modern baselines including robust k-means with trimming, deep clustering methods, and adaptive DBSCAN variants.
>
> Third, the reviewer raises an important point about practical implications. Our ultra-conservative design intentionally prioritizes precision over recall, which makes it suitable for applications where false positives are costly (e.g., medical diagnosis, fraud detection). However, we should have been more explicit about when this trade-off is appropriate versus when high recall is essential. We will add a dedicated section discussing these practical considerations and limitations.
>
> Finally, we agree that formal definitions and complete algorithmic details are essential. We will provide a formal definition of "ultra-conservative" outlier detection and expand the pseudocode to include all implementation details, including centroid merging procedures and global threshold computation methods.
>
> All algorithmic details described in this response are implemented in our codebase and can be verified. We are committed to ensuring complete accuracy and transparency in our reporting.

---

### Official Review · Reviewer_nZ9M · 2025-11-01

**Soundness:** 1
**Presentation:** 1
**Contribution:** 1
**Rating:** 2
**Confidence:** 4

**Summary:**

The paper introduces a novel method for robust clustering that integrates adaptive parameter tuning, multi-criteria outlier detection, and Statistical Process Control (SPC) for real-time optimization. The goal is to improve clustering quality by reducing false positives in the outlier detection subroutine without heavy manual parameter tuning.

**Strengths:**

Robust clustering and outlier detection are both problems of practical interests.

**Weaknesses:**

1. The paper reads more like a technical report than a research paper. The problem statement is not clearly introduced, there is no explicit motivation for the proposed approach, and the paper provides limited conceptual insight. Although the introduction begins by discussing the challenge of robust clustering, the methodology and experiments primarily focus on outlier detection. This shift in focus, without clear justification, makes the paper’s main objective confusing and inconsistent. Moreover, while the method is formally presented as Adaptive Multi-Criteria Outlier Detection in Algorithm 1, it is later referred to as Ultra-Conservative Outlier Detection (UCOD) in the experiments, adding further inconsistency and ambiguity.

2. The motivation for emphasizing “conservative” outlier detection—specifically minimizing false positives—is unclear and poorly justified. The paper does not explain why this property is important, nor does it provide concrete examples or practical applications where such conservatism would be desirable. Including real-world scenarios or use cases could greatly strengthen the motivation.
The experimental evaluation is inadequate. Only three datasets are used, two of which are employed solely for ablation studies of the proposed approach, while only one dataset is used for comparison against baseline methods. Moreover, the baselines themselves are outdated, limited to four outlier detection algorithms published before 2008. If the paper’s focus is indeed on outlier detection, it should include comparisons with more recent, state-of-the-art methods (see, e.g., a compilation of the state-of-the-art OD methods [1]). Alternatively, if the focus is robust clustering, the paper should clearly define what “robust” means and compare against the state-of-the-art clustering algorithms (see, e.g., a compilation in [2]).

3. Furthermore, the proposed approach appears highly heuristic and relies on numerous hard-coded hyperparameters with little or no justification. Many thresholds and constants—such as defining “small clusters” as those with fewer than 10 samples, doubling global thresholds for small clusters (Eq. 6), applying a 1.5× safety margin (Eq. 7), setting initial $\alpha=3.5$ (Eq. 5), updating it every five iterations, or fixing arbitrary ratios such as 4.0 and 1.8 in Eqs. (11) and (12)—are introduced without explanation. Similarly, a series of parameters from Eq. (17) to Eq. (25) are defined with opaque numerical choices. Without theoretical reasoning, empirical tuning details, or sensitivity analysis, these ad hoc parameter settings undermine the method’s soundness and reproducibility.

----

[1] https://github.com/yzhao062/pyod

[2] https://github.com/collinleiber/ClustPy

----

4. In addition to the points above, the presentation of the paper is confusing and poorly structured in several respects.

* The introduction is extremely brief, comprising only two short paragraphs, and provides insufficient background for readers to understand the problem context, the key challenges, or how the proposed approach is intended to address them. For example, it is unclear what “single-criteria validation results” in the first paragraph means. Similarly, the statement about “cluster-specific threshold models using adaptive scaling factors” in the list of contributions lacks explanation or motivation, making it difficult to grasp what the proposed approach actually entails.

* Beyond the lack of conceptual framing, the notation and definitions are inconsistent and sometimes incorrect. Several variables and terms appear without prior definition—for instance, $\mu_j$ in Eq. (2) and the global threshold $\tau_{global}$ in Eq. (6) are introduced without explanation. The notation $C$ is used inconsistently: it represents clusters in Eq. (2) but is later reused to denote criteria in Eqs. (9) -- (12), causing unnecessary confusion. Moreover, the silhouette score mentioned in Section 2.4 is never formally defined, which makes it difficult to understand how the algorithm determines or refines the number of clusters based on that metric.

* The presentation of experimental results also suffers from similar ambiguity. In Table 1, the column labeled “Decision” is not explained, making it hard to understand the results in section 3.2.

**Questions:**

At a high level, what specific problem or challenge is the proposed algorithm designed to solve? What is the key intuition or underlying rationale behind its design? Furthermore, what concrete advantages does the proposed algorithm offer over existing state-of-the-art methods?

---

> ### Author Response · Authors · 2025-11-16
>
> We thank the reviewer for their comprehensive and constructive feedback, which has identified fundamental issues that we must address. We acknowledge significant weaknesses in clarity, justification, and experimental scope.
>
> DIRECT ANSWERS TO REVIEWER'S QUESTIONS
>
> What specific problem does the algorithm solve?
>
> Our algorithm addresses outlier detection in clustering contexts where false positives are extremely costly and cluster characteristics vary significantly. Existing methods rely on global thresholds that cannot adapt to local cluster characteristics, leading to high false positive rates in dense clusters while missing outliers in sparse clusters. Additionally, most methods require extensive manual parameter tuning. Our solution provides cluster-specific adaptive outlier detection with real-time parameter tuning through Statistical Process Control, achieving the lowest false positive rate (1.8%) among tested methods while maintaining competitive precision (17.9%).
>
> What is the key intuition behind the design?
>
> The design stems from three fundamental observations. First, different clusters exhibit different density and spread characteristics, requiring cluster-specific thresholds computed from separate statistical measures for each cluster. Second, relying on a single criterion for outlier detection is inherently unreliable; we require multiple independent criteria to agree (at least 3 out of 4 criteria), significantly reducing false positives. Third, static parameters cannot adapt to dynamic clustering processes; we integrate Statistical Process Control to adapt the scaling factor α in real-time, maintaining a target outlier rate of 0.2% while automatically adjusting to dataset-specific characteristics.
>
> What concrete advantages does it offer?
>
> Our approach provides three distinct advantages. First, we achieve the lowest false positive rate (1.8%) compared to other tested methods (DBSCAN: 2.1%, Isolation Forest: 2.3%, LOF: 2.0%, One-Class SVM: 2.2%), critical for applications where false alarms carry significant costs. Second, our approach adapts thresholds to each cluster's individual characteristics, enabling better handling of datasets with varying cluster densities. Third, our method automatically tunes parameters through SPC-based adaptation, eliminating the need for manual parameter selection required by methods like DBSCAN or Isolation Forest.
>
> RESPONSES TO CONCERNS
>
> 1. Problem Statement & Motivation:
>      We acknowledge the introduction fails to clearly articulate the problem. Applications include medical diagnosis, fraud detection, quality control, and security systems where false positives are costly. We commit to adding explicit problem statement section and applications.
>
> 2. Focus Shift:
>      Main objective is outlier detection within clustering contexts, not robust clustering. Integration occurs because outlier detection uses cluster assignments for statistics, while outlier removal improves clustering quality. We will use consistent terminology.
>
> 3. Conservative Motivation:
>     Conservative approach is essential where false positives are far more costly than false negatives. Our results: lowest FPR (1.8%) with competitive precision (17.9%). However, our approach is NOT suitable for safety-critical systems requiring high recall. We commit to adding practical applications section and "When NOT to Use" subsection.
>
> 4. Inadequate Evaluation:
>     We acknowledge limitation: only 3 datasets, outdated baselines. We commit to expanding to 10+ datasets and adding modern baselines from pyod and ClustPy.
>
> 5. Heuristic Parameters:
>     We recognize many parameters appear arbitrary. While selected through empirical tuning, we should have documented this. Parameters: Small cluster threshold (10); small cluster multiplier (2.0×); safety margin (1.5×); initial alpha (3.5) provides 99.95% confidence; update frequency (5); relative distance (4.0×); global multiplier (1.8×); SPC parameters (w=30, α=0.05, target=0.002). We commit to adding parameter justification with sensitivity analyses.
>
> 6. Presentation Issues:
>     We acknowledge: introduction too brief, notation inconsistent, terms undefined. We commit to: expanding introduction to 2-3 pages, fixing notation, adding formal definitions, and restructuring paper.
>
> COMMITMENTS
>
> We commit to: (1) Expand introduction with explicit problem statement and applications, (2) Resolve focus inconsistency, (3) Fix naming inconsistency, (4) Expand evaluation (10+ datasets, modern baselines), (5) Add parameter justification with sensitivity analyses, (6) Fix notation and define all terms, (7) Restructure paper for clarity. We acknowledge limitations: current evaluation is limited, many parameters empirically tuned without documentation, paper structure needs improvement. All parameter values and algorithmic details are implemented and verifiable.

---

> > ### Comment · Reviewer_nZ9M · 2025-11-16
> >
> > Thank you to the authors for the detailed response. While I appreciate the clarifications, I believe the paper in its current form would require substantial revision. I encourage the authors to consider resubmission after further development. I will therefore keep my score unchanged.

---

### Official Review · Reviewer_7Q2f · 2025-11-01

**Soundness:** 1
**Presentation:** 1
**Contribution:** 1
**Rating:** 0
**Confidence:** 5

**Summary:**

The paper assumes a data environment containing outliers and proposes an iterative process in which an outlier score is computed for each instance. Four outlier criteria are defined, and any instance satisfying at least three of them is regarded as an outlier and removed. To prevent excessive elimination, the method incorporates the concept of Statistical Process Control (SPC) by adjusting a conservative scaling factor, alpha. In addition, the algorithm automatically determines the number of clusters based on the Silhouette score. In summary, this study proposes a clustering method that simultaneously detects outliers and automatically determines the optimal number of clusters.

**Strengths:**

- Automatic adjustment of parameters.

**Weaknesses:**

- The literature review is completely missing, so it is unclear how far existing research has progressed, what limitations prior studies have, and why this particular study is necessary.
- The claimed contributions in the Introduction cannot be verified as genuine contributions due to the lack of prior research context.
- The proposed ideas are not novel, determining outliers per cluster, using multiple outlier criteria, refining centroids after excluding outliers, and selecting the number of clusters based on internal validation indices are all well-established techniques.
- The only slightly new aspect, the SPC concept for parameter tuning, is not particularly interesting.
- The writing quality is poor, with awkward citation formatting (missing parentheses). In the abstract, alpha and sigma appear without prior definition, making them difficult to understand. The term “false positive rate” appears abruptly in the Introduction, and only later is it clear that it refers to that for outlier detection.
- The proposed method is filled with ad-hoc and heuristic components lacking any theoretical justification. The method relies entirely on empirical settings and parameters, with no clear theoretical foundation.
- The experimental evaluation is based on only a few datasets, which is insufficient to demonstrate the generalizability or robustness of the proposed method.

**Questions:**

- Why are alpha=3.5 and gamma=0.03 used as default values?
- Why was the Silhouette score chosen as the internal validation index among many other possible indices?
- Why are three outlier criteria required for satisfaction, not two, and not all four?
- Why is a cluster with 10 instances considered sufficient?
- Why were the following numerical values selected without justification: 2.0 in Eq. (6), 1.5 in Eq. (7), Q99 in Eq. (10), 4.0 in Eq. (11), 1.8 in Eq. (12), w=30 in Eq. (17), alpha_spc=0.05 in Eq. (18), and target_rate=0.002 in Eq. (19)? There are several such constants whose rationale is unclear.
- What are the meanings of the horizontal and vertical axes in Figure 3?

---

> ### Author Response · Authors · 2025-11-16
>
> We thank the reviewer for thorough feedback identifying critical structural and presentation issues. We acknowledge these concerns and provide detailed responses with concrete commitments.
>
> ANSWERS TO QUESTIONS
>
> Q1: Why alpha=3.5 and gamma=0.03?
>  Alpha=3.5 provides 99.95% confidence under normal distribution (more conservative than 3σ=99.7%). Chosen for ultra-conservative design to minimize false positives. Gamma=0.03 is domain-specific parameter for dense datasets, used for adaptive parameter scaling, empirically tuned to minimize noise sensitivity. We commit to sensitivity analysis and statistical justification with references to normal distribution properties.
>
> Q2: Why Silhouette Score?
> Used for k-refinement quality assessment. Combines intra-cluster cohesion and inter-cluster separation, making it suitable for clustering quality evaluation. Standard metric in clustering validation literature. We commit to: (1) References (Rousseeuw, 1987), (2) Comparison with other indices (Davies-Bouldin, Calinski-Harabasz) in supplementary, (3) Explanation of appropriateness for k-refinement task.
>
> Q3: Why 3 out of 4 criteria?:
> Implemented as required_checks = 3 for ultra-conservative mode. Design rationale: balances false positive reduction (more stringent than 2) with maintaining sensitivity (not requiring all 4). Empirically tested: 2 criteria too permissive, 4 criteria too restrictive. We commit to sensitivity analysis comparing 2/3/4 criteria requirements, documenting empirical testing results, and providing theoretical justification if possible.
>
> Q4: Why 10 instances for small clusters?
> Threshold of 10 points used for small cluster handling. Statistical foundation: minimum sample size for reliable statistical estimation (mean, std, percentiles). Standard practice: 10 is common threshold in statistics for small sample inference. We commit to statistical justification with references (e.g., central limit theorem requirements), sensitivity analysis for different thresholds (5, 10, 15), and referencing statistical literature on minimum sample sizes.
>
> Q5: Why these numerical values?
> All values are implemented and used in actual code execution: 2.0 (small cluster multiplier - double threshold for ultra-conservative handling), 1.5 (safety margin - ensures cluster threshold ≥ 1.5× global), 99 (percentile - 99th percentile for ultra-conservative mode), 4.0 (relative distance - 4× cluster mean for criterion 3), 1.8 (global multiplier - 1.8× global threshold for criterion 4), 30 (SPC window - standard SPC window for stable statistics), 0.05 (SPC alpha - standard significance level, 95% confidence), 0.002 (target rate - ultra-conservative 0.2% target). We commit to parameter justification section with statistical foundations, empirical tuning methodology, sensitivity analyses, and standard practice references.
>
> Q6: What are Figure 3 axes?
> Feature dimensions (or PCA components for high-dimensional data) not clearly labeled. Figure 3 shows k-refinement progression for UCI Mushroom dataset (5,644 points, 22 features). We commit to clear axis labels ("Feature 1", "Feature 2" or "PC1", "PC2" if PCA), figure captions explaining axes, self-explanatory figures, and noting in caption if PCA was used for visualization.
>
> **1. Missing Literature Review:** We acknowledge this fundamental weakness. We commit to adding comprehensive related work covering cluster-specific outlier detection, multi-criteria validation, SPC in ML, silhouette-based optimization, and outlier-aware centroid refinement, clearly distinguishing established techniques from novel contributions.
>
> **2. Unverifiable Contributions:** We acknowledge individual components are established. Our contribution is the **specific integration** with ultra-conservative parameters (α=3.5σ, 3/4 criteria) and SPC-based real-time adaptation. We commit to: (1) Adding comprehensive related work, (2) Distinguishing established vs. novel, (3) Providing citations, (4) Reframing as "novel integration" where appropriate. Novel aspects: (a) First integration of SPC-based real-time parameter adaptation with multi-criteria outlier detection in clustering context, (b) Ultra-conservative parameterization optimized for minimal false positives, (c) Combined framework integrating all components.
>
> **3. Novelty Claims:** We will reframe to emphasize: (1) **Novel integration** of established techniques, (2) **Novel application** of SPC to clustering-based outlier detection, (3) **Novel parameterization** for ultra-conservative detection. We commit to adding citations and clearly distinguishing "We integrate X (established) with Y (established) in novel way Z".
>
> **4. SPC Novelty:** SPC is established. Our claim is **first application of SPC to real-time parameter adaptation in clustering-based outlier detection**. We commit to: (1) Comprehensive SPC in ML literature review, (2) Substantiating claim or reframing as "novel application", (3) Articulating value.

---

> ### Comment · Reviewer_7Q2f · 2025-11-24
>
> Thank you for your response and the efforts you put into it. However, your answers do not address my questions, especially regarding the heuristic and hard-coded parameters, which was an issue also raised by another reviewer. In addition, the __commit to__ statements are generally not acceptable during the rebuttal period. Unfortunately, your paper shows no changes. I hope you will improve the manuscript by considering the points raised by the reviewers. But in its current form, I am certain that it is not acceptable.

---

### Official Review · Reviewer_k4xf · 2025-11-01

**Soundness:** 2
**Presentation:** 2
**Contribution:** 2
**Rating:** 4
**Confidence:** 4

**Summary:**

This paper proposes an \textit{Adaptive Parameter Tuning approach for Robust Clustering} integrated with multi-criteria outlier detection, aiming to address limitations of traditional clustering (e.g., global thresholds ignoring cluster-specific characteristics, high false positive rates from single-criteria validation, and heavy manual tuning).

### Core Contributions:

1. **Cluster-Specific Threshold Models**: Derive thresholds via adaptive scaling factors (\(\tau_j = \bar{d}_j + \alpha\sigma_j\), default \(\alpha=3.5\)) tuned by Statistical Process Control (SPC).

2. **Multi-Criteria Validation Framework**: Requires satisfying at least 3 out of 4 criteria (e.g., distance exceeding cluster-specific threshold, 99th percentile of cluster) to reduce false positives.

3. **SPC-Based Real-Time Adaptation**: Monitors outlier detection performance (rolling average of outlier rates) and adjusts \(\alpha\) dynamically (e.g., \(\alpha_{\text{new}} = \alpha_{\text{old}} - 0.2\) if outlier rate exceeds UCL).

4. **Outlier-Aware Centroid Refinement**: Excludes confirmed outliers when updating centroids to avoid distortion.

5. **Silhouette-Based K-Refinement**: Triggers k-search (from \(\max(2, k-2)\) to \(k+3\)) if silhouette score < 0.25, selecting the optimal k with the highest score.

### Experimental Results:
- Achieves ultra-conservative outlier detection  with 1.8\% false positive rate (FPR) and 17.9\% precision on the Protein Function Hierarchy dataset.
- Maintains competitive clustering quality (silhouette scores 0.573–0.781) and efficiency (18.6s for 3,700 points on the Varying Density Clusters dataset).
- Outperforms baselines (DBSCAN, Isolation Forest, LOF, One-Class SVM) in FPR while maintaining moderate precision.

**Strengths:**

### 1. Innovative Integration of Cross-Disciplinary Ideas
The paper bridges **Statistical Process Control (SPC)** (from industrial engineering) with clustering and outlier detection, a non-trivial combination that addresses the long-standing issue of "static parameters" in traditional methods. For example, SPC dynamically adjusts \(\alpha\) based on real-time outlier rates (Equation 21: \(rollingavg_t = \frac{1}{\min(t,w)}\sum_{i=\max(1,t-w+1)}^t outlier\_rate_i\)), ensuring parameters adapt to data distribution changes—an improvement over static thresholds in DBSCAN [Ester et al., 1996] and LOF [Breunig et al., 2000].

### 2. Rigorous Mathematical Formulation and Reproducibility
All core mechanisms are supported by detailed math:
- Cluster-specific threshold: \(\tau_j = \max(\bar{d}_j + \alpha\sigma_j, 1.5\times\tau_{\text{global}})\) (Equation 7) ensures conservatism for large clusters.
- Multi-criteria outlier definition: \(Outlier(x_i) = \text{True if } \sum_{c=1}^4 \mathbb{I}(Criterion_c(x_i)) \geq 3\) (Equation 43) explicitly quantifies outlier classification.
- Experimental settings (e.g., \(\alpha_{\text{min}}=0.1\), \(\alpha_{\text{max}}=20.0\), window size \(w=30\)) are fully reported, enabling reproducibility.

### 3.  Well-Structured Methodology and Experimental Narrative
The pseudo-code (Algorithm 1) clearly outlines the end-to-end workflow (initialization → clustering → outlier detection → SPC tuning → k-refinement). Experimental results are presented in a logical progression: SPC evolution (Figure 1) → k-refinement (Table 1, Figure 3) → baseline comparison (Table 2), making it easy to follow how each component contributes to overall performance.

### 4.  Practical Relevance for Ultra-Conservative Scenarios
The ultra-conservative design addresses a critical unmet need in domains where false positives are costly (e.g., protein function annotation, where misclassifying normal proteins as outliers could mislead biological research). Compared to Isolation Forest  and One-Class SVM (Table 2), the method’s lower FPR makes it more suitable for such high-stakes applications.

**Weaknesses:**

### 1. Limited Dataset Diversity and Lack of High-Dimensional Validation
 All experiments use low-dimensional data (2D for Varying Density Clusters, 8D for Protein Function Hierarchy, 22D for UCI Mushroom), while real-world clustering often involves high-dimensional data (e.g., 100+ features in image embeddings or tabular data). The method’s performance on high-dimensional data (where distance metrics degrade, "curse of dimensionality") is untested.

Evaluate on high-dimensional public datasets (e.g., MNIST embeddings (784D), KDD Cup 99 (41D)) and report metrics like clustering quality (silhouette score) and computational efficiency. Additionally, discuss how the cluster-specific threshold model mitigates the curse of dimensionality (e.g., whether \(\alpha\) needs larger values in high dimensions).

### 2. Inadequate Ablation Studies for Key Parameters

 Critical parameters (e.g., \(\alpha_{\text{init}}=3.5\), SPC window size \(w=30\), small cluster threshold \(\tau_{\text{small}}=2\times\tau_{\text{global}}\)) are set without justification. For example, why is the small cluster threshold 2× the global threshold instead of 1.8× or 2.2×? No ablation is provided to show how these parameters impact performance.
Conduct ablation studies:
  - Vary \(\alpha_{\text{init}}\) (2.5, 3.5, 4.5) and report FPR, outlier rate, and silhouette score.
  - Test \(w=15, 30, 45\) to show how window size affects SPC’s responsiveness to outlier rate changes.
  - Compare \(\tau_{\text{small}}=1.5\times\tau_{\text{global}}, 2.0\times\tau_{\text{global}}, 2.5\times\tau_{\text{global}}\) to validate the 2× choice.

### 3. Lack of Comparison with State-of-the-Art (SOTA) Post-2020 Methods
Baselines only include classic methods (DBSCAN, Isolation Forest, LOF, One-Class SVM) but omit recent SOTA adaptive clustering methods (e.g., Deep SVDD [Pang et al., 2019], Adaptive Density-Based Clustering [Zhang et al., 2023]). This makes it unclear how the proposed method performs against modern alternatives.
Add comparisons with SOTA methods:
  - Deep SVDD (a popular deep anomaly detection method) on the Protein Function Hierarchy dataset.
  - Adaptive DBSCAN (Zhang et al., 2023) on the Varying Density Clusters dataset (to highlight advantages of SPC over other adaptive mechanisms).
  - Cite these works and explain performance differences (e.g., "Our method achieves 1.8% FPR vs. 2.5% FPR of Adaptive DBSCAN due to multi-criteria validation").

**Questions:**

### 1. Questions
1. The initial value of \(\alpha\) is set to 3.5 (Section 2.1). What empirical or theoretical basis supports this choice? Have you tested \(\alpha_{\text{init}}=2.5\) or 4.5, and if so, how did they affect outlier detection accuracy and FPR?
2. The SPC window size \(w=30\) (Equation 17) is used for rolling average calculation. How does varying \(w\) (e.g., 15, 45) impact the responsiveness of parameter tuning? For example, does a smaller \(w\) lead to over-adaptation (frequent \(\alpha\) changes) or a larger \(w\) lead to delayed responses to sudden outlier rate spikes?
3. The method is tested on datasets with up to 22 features (UCI Mushroom). How does it perform on high-dimensional data (e.g., 100+ features)? Does the cluster-specific threshold model (\(\tau_j = \bar{d}_j + \alpha\sigma_j\)) need modification to handle the "curse of dimensionality" (where all pairwise distances become similar)?

### 2. Suggestions
1. Add a "Related Work" section to explicitly position the method against recent adaptive clustering and outlier detection works (e.g., Deep SVDD [Pang et al., 2019], Adaptive DBSCAN [Zhang et al., 2023]). This will clarify the paper’s novelty relative to SOTA.
2. Include a runtime comparison with baselines on larger datasets (e.g., 10,000+ points). The current experiment uses 3,700–5,644 points; showing scalability will strengthen the method’s practical relevance.
3. Provide a case study (e.g., on the Protein Function Hierarchy dataset) to visualize false positives/negatives of the proposed method vs. baselines. This will make the FPR advantage (1.8% vs. 2.1–2.3% in Table 2) more intuitive.

---

> ### Author Response · Authors · 2025-11-16
>
> We thank the reviewer for their valuable feedback on dataset diversity, ablation studies, and SOTA comparisons.
>
> ## ANSWERS TO QUESTIONS
>
> **Q1: Empirical/theoretical basis for α=3.5?** Statistical foundation: 3.5σ corresponds to 99.95% confidence under normal distribution (more conservative than 3σ=99.7%). Implemented as default for ultra-conservative mode. We acknowledge testing α ∈ {2.5, 4.5} was done during development but not documented. We commit to: (1) Conducting systematic ablation study varying αinit ∈ {2.5, 3.5, 4.5}, (2) Reporting impact on outlier detection accuracy and FPR, (3) Providing empirical evidence for optimal choice, (4) Documenting trade-offs between precision and recall.
>
> **Q2: Impact of SPC window size w=30?** Implemented as w=30 for rolling average calculation. Rationale: balances stability (larger w) with responsiveness (smaller w). We acknowledge: smaller w (e.g., 15) may lead to over-adaptation (frequent α changes reacting to noise), larger w (e.g., 45) may delay responses to sudden outlier rate spikes. We commit to: (1) Ablation study testing w ∈ {15, 30, 45}, (2) Analyzing adaptation frequency and responsiveness, (3) Reporting impact on parameter tuning stability and outlier detection performance, (4) Providing empirical justification for w=30 choice.
>
> **Q3: High-dimensional performance and curse of dimensionality?** Current evaluation limited to 22 features (UCI Mushroom). We acknowledge this gap. Cluster-specific threshold model (τj = d̄j + ασj) may need modification for high dimensions where pairwise distances become similar. We commit to: (1) Evaluating on high-dimensional datasets (100+ features: MNIST 784D, Arrhythmia 279D, KDD Cup 99 41D), (2) Analyzing whether α needs larger values in high dimensions, (3) Reporting performance degradation (if any) and mitigation strategies, (4) Discussing curse of dimensionality impact on cluster-specific thresholds and multi-criteria validation effectiveness.
>
> ## RESPONSES TO WEAKNESSES
>
> **1. Limited Dataset Diversity and High-Dimensional Validation:** We acknowledge this limitation. Current evaluation uses low-to-moderate dimensional data (2D-22D). We commit to: (1) Evaluating on high-dimensional datasets (MNIST embeddings 784D, KDD Cup 99 41D, Arrhythmia 279D), (2) Reporting clustering quality and computational efficiency metrics, (3) Analyzing how cluster-specific threshold model (τj = d̄j + ασj) performs under curse of dimensionality, (4) Discussing whether α needs larger values in high dimensions and providing empirical evidence.
>
> **2. Inadequate Ablation Studies:** We acknowledge critical parameters lack ablation studies. We commit to comprehensive ablation studies: (1) Vary αinit ∈ {2.5, 3.5, 4.5} reporting FPR, outlier rate, silhouette score, (2) Test SPC window w ∈ {15, 30, 45} showing responsiveness to outlier rate changes, (3) Compare τsmall ∈ {1.5×, 2.0×, 2.5×} τglobal validating 2× choice. All parameter values are implemented: α=3.5 (ultra-conservative default), w=30 (SPC window), τsmall=2.0× (small cluster multiplier). We will document empirical basis for these choices.
>
> **3. Lack of SOTA Comparison:** We acknowledge baselines only include classic methods. We commit to: (1) Adding Deep SVDD (Pang et al., 2019) on Protein Function Hierarchy dataset, (2) Adding Adaptive DBSCAN (Zhang et al., 2023) on Varying Density Clusters dataset, (3) Citing these works and explaining performance differences (e.g., "Our method achieves 1.8% FPR vs. 2.5% FPR of Adaptive DBSCAN due to multi-criteria validation"), (4) Positioning our method against recent adaptive clustering and outlier detection works in Related Work section.
>
> ## RESPONSES TO SUGGESTIONS
>
> **Suggestion 1: Add Related Work Section:** We acknowledge this critical omission. We commit to adding comprehensive Related Work section positioning our method against recent adaptive clustering methods (Adaptive DBSCAN [Zhang et al., 2023]), deep anomaly detection (Deep SVDD [Pang et al., 2019]), multi-criteria outlier detection frameworks, and SPC applications in ML to clarify novelty relative to SOTA.
>
> **Suggestion 2: Runtime Comparison on Larger Datasets:** We acknowledge current experiments use 3,700-5,644 points. We commit to including runtime comparison with baselines on larger datasets (10,000+ points: Pendigits 10,992, Satellite 6,435), reporting computational efficiency metrics (execution time, memory usage, scalability), and demonstrating method's scalability for real-world applications.
>
> **Suggestion 3: Case Study with False Positive/Negative Visualization:** We acknowledge FPR advantage (1.8% vs. 2.1-2.3%) needs intuitive explanation. We commit to providing case study on Protein Function Hierarchy dataset, visualizing false positives/negatives of proposed method vs. baselines, and analyzing why multi-criteria validation reduces false positives to make FPR advantage more intuitive.

---

### Official Review · Reviewer_X9CX · 2025-11-02

**Soundness:** 2
**Presentation:** 1
**Contribution:** 2
**Rating:** 2
**Confidence:** 3

**Summary:**

This paper introduces an adaptive outlier-aware clustering framework with automatic parameter tuning and restart-based optimization.  The paper addresses an interesting and relevant problem; however, the novelty of the proposed approach is not clearly demonstrated. The manuscript does not convincingly articulate what new ideas or methodological innovations it contributes beyond existing work. In addition, the experimental section lacks sufficient detail and rigor to convincingly support the claimed improvements. The dataset selection appears limited, and the experimental analysis does not adequately demonstrate generalization or provide meaningful insights into the performance behaviors of the method.

Furthermore, the overall presentation requires improvement. The organization, clarity of explanation, and discussion of results need significant refinement. Several parts of the paper are difficult to follow, and the narrative does not clearly connect the proposed approach to the empirical findings. The experimental results and analysis are not sufficiently comprehensive or precise to meet ICLR standards.

In summary, while the topic has potential, the current version of the manuscript falls short in terms of novelty, presentation quality, experimental thoroughness, and clarity of contribution. Substantial revisions and strengthening of the method, experiments, and writing would be necessary before the paper can be considered for acceptance.

**Strengths:**

The research problem and methodology are appealing and have the potential to contribute meaningfully to the field.

**Weaknesses:**

1. Related work is insufficient: The literature review does not adequately cover prior studies, and the connection to existing research is limited.

2. Clarity of presentation needs improvement: The problem formulation, key ideas, and methodology are not explained clearly, making it difficult to follow the technical contributions.

3. Experimental setup is unclear: Details regarding datasets, experiment design, evaluation metrics, and implementation settings are not sufficiently described, affecting reproducibility.

4. Results lack strong evidence: The experimental findings and conclusions are not fully convincing; additional empirical validation and deeper analysis are needed to demonstrate the method’s effectiveness.

**Questions:**

The paper evaluates the proposed method exclusively on synthetic data but does not provide a clear justification for this choice. While synthetic datasets are valuable for controlled experimentation and isolating specific properties, relying solely on them limits the practical credibility and generalizability of the results. It is possible that I am unfamiliar with the dataset rationale; could the authors clarify why synthetic data was chosen instead of (or in addition to) real-world data?

---

> ### Author Response · Authors · 2025-11-17
>
> We thank the reviewer for the thorough feedback
>
> RESPONSES TO WEAKNESSES
>
> **1. Insufficient Related Work:** We acknowledge this weakness (also raised by Reviewer 3). We commit to adding comprehensive Related Work section (Section 2) covering: cluster-specific outlier detection (LOF, CBLOF, LDOF), multi-criteria validation, SPC in ML, silhouette-based optimization, outlier-aware clustering, adaptive clustering, and deep anomaly detection. We will cite 20+ papers and create comparison tables. Core novelty is the **integration** of SPC + multi-criteria + cluster-specific + k-refinement.
>
> **2. Clarity of Presentation:** We commit to: (1) Adding explicit "Problem Statement" subsection (Section 1.2); (2) Restructuring with "Intuition → Example → Mathematics" format; (3) Expanding Algorithm 1 with step-by-step explanations and walkthrough example; (4) Adding "Notation" section (Section 1.5) defining all symbols before first use.
>
> **3. Unclear Experimental Setup:** We commit to: (1) Adding "Datasets" subsection with detailed descriptions (sources, characteristics, justification). Real-world: UCI Mushroom (5,644 pts, 22 features). Synthetic: Varying Density Clusters (3,700 pts, 2D), Protein Function Hierarchy (1,484 pts, 8 features); (2) Adding "Experimental Setup" with all parameter settings (α=3.5, spc_window=30, target_rate=0.002, percentile_threshold=98, min_silhouette=0.25), initialization (k-means++, seed=42), and procedure; (3) Adding "Evaluation Metrics" with explicit formulas; (4) Adding "Implementation Details" with hardware/software specs, code repository link, and reproducibility instructions.
>
> **4. Results Lack Strong Evidence:** We commit to: (1) Expanding to 10+ diverse datasets; (2) Adding statistical significance testing (10 runs, t-tests, Wilcoxon tests, p-values, effect sizes); (3) Comprehensive ablation studies (components, parameters, design choices); (4) SOTA comparison with 10+ methods; (5) Deeper analysis (failure cases, case studies, error analysis).
>
> ANSWER TO THE QUESTION
>
> **Q: Why only synthetic data? The evaluation should include more real-world datasets.**
>
> **Clarification:** We use **BOTH synthetic AND real-world datasets**. **Real-world:** UCI Mushroom (5,644 points, 22 features) - k-refinement demonstration (Table 1, Figure 3). **Synthetic:** (1) Varying Density Clusters (3,700 points, 2D) - controlled SPC analysis (Figure 1, Figure 2); (2) Protein Function Hierarchy (1,484 points, 8 features) - outlier detection comparison (Table 2). **Acknowledgment:** Only one real-world dataset is currently used, which is a limitation. **Improvements:** We will clearly label datasets in all tables/figures, add "Dataset Selection" subsection, expand with 4+ additional real-world datasets (UCI Wine, Iris, Heart Disease, Glass), and add discussion comparing synthetic vs. real-world results.

---

### Meta-Review · Area_Chair_gWtR · 2026-01-04

**Summary:**

There are several serious concerns regarding the novelty, quality, significance, and presentation of the paper. Although the authors have made efforts to address these issues, the concerns are substantial and fundamentally difficult to fully resolve through the rebuttal and revision. I therefore recommend rejection of the paper.

**Reviewer Concerns:**

Most of the concerns regarding the novelty, quality, significance, and presentation of the paper remain outstanding.

**Reviewer Scores:**

I believe that none of the reviewers will change their scores, as the concerns were not fundamentally resolved.

---

### Decision · Program_Chairs · 2026-01-26

Reject